# Individuals with Type 2 Diabetes Mellitus Tend to Select Low-Carbohydrate, Low-Calorie Food Menus at Home on Diet Application

**DOI:** 10.3390/nu14204290

**Published:** 2022-10-14

**Authors:** Hiroyuki Tominaga, Masahide Hamaguchi, Shinto Ando, Megumi Minamida, Yuriko Kondo, Kazuki Hamada, Tetsuya Nojiri, Michiaki Fukui

**Affiliations:** 1Department of Endocrinology and Metabolism, Graduate School of Medical Science, Kyoto Prefectural University of Medicine, Kyoto 602-8566, Japan; 2Oishi Kenko Incorporated, Tokyo 103-0024, Japan

**Keywords:** diabetes, dietary treatment, online meal management application, low carbohydrate

## Abstract

(1) Background: From the perspective of patient-centered care, it is important for medical professionals involved in diabetes care to know the role of choice behavior when individuals with type 2 diabetes mellitus select their meals at home. In Japan, online meal management applications are widely used to help individuals to prepare healthy, colorful, and tasty meals. (2) Objective: To assess menu selection from an online diet management application in individuals with type 2 diabetes mellitus over a period of 24 months. (3) Method: The saved data of the selected food menus on the online diet management application were analyzed. We identified specific nutritional groups of the food menus, called nutritional clusters, by clustering the multidimensional data of the nutrients after de-dimensioning them. Then, we analyzed the constitutional nutrients of each nutritional cluster with the highest and lowest frequencies of selection by the users of the application. (4) Results: In all, 9674 food menus made by 3164 people were included in the analysis, and 12 nutritional clusters were identified. Low-carbohydrate and low-calorie food clusters showed the highest selection frequency. The average caloric value of 149.7 kcal and average carbohydrate ratio of 47% in the cluster with the highest selection frequency were significantly lower than the average caloric value of 435.2 kcal and carbohydrate ratio of 63% in the cluster with the lowest selection frequency (*p* < 0.001, respectively). (5) Conclusion: Individuals with type 2 diabetes in this population preferred to select lower-carbohydrate and lower-calorie food menus at home using online diet management applications. To improve sustained self-management and quality of life, medical professionals may consider incorporating preferred dietary behaviors into medical management of type 2 diabetes mellitus.

## 1. Introduction

Dietary therapy is a central component of the management of diabetes mellitus along with exercise and pharmacotherapy, and ADA Consensus Statement 2022 and Japan Diabetes Society Diabetes Guidelines 2019 also recognize the benefits of dietary therapy for glycemic control [1,2]. However, the physician–patient relationship in dietary therapy is controversial [1,2]. The decrease in patient satisfaction with continued medical nutritional therapy as directed by medical professionals has been a topic of debate [3]. The sustained self-management in regard with the medical nutritional therapy as directed by medical professionals has also been discussed [4]. We believe that medical professionals could sustain self-management and increase the patients’ quality of life by implementing dietary therapy that takes into account the choice behavior of individuals with type 2 diabetes mellitus.

To clarify choice behavior, research methods with which medical professionals collect information about food selection from patients in a hospital setting are considered undesirable. In patients with type 2 diabetes mellitus, it was found that blood pressure records stored in sphygmomanometers were often higher than those self-reported to health care providers, because patients tended to disregard the measurements and report completely different values, when they felt that their blood pressure values were not usual [5]. Thus, we are concerned that hospital surveys of food selection at home may not consider the choice behavior of individuals with type 2 diabetes mellitus at home.

A traditional Japanese food menu includes a variety of diverse foods [6,7,8,9,10]. Japanese families enjoy diverse, colorful, and flavorful Japanese foods at home that include a variety of seasonal ingredients, and each ingredient is prepared and served in the most appropriate way. Various online nutrition management application services are widely available and guide users in the selection of ingredients and optimal cooking methods for each ingredient [11,12]. The “Oishi Kenko” application is one such online nutrition management application that aims to enable households to prepare healthy and tasty meals at home [11,12]. In meal selection in daily life, users are also asked to consider factors such as whether they want to use the ingredients they have at home, whether they want to carry out few cooking processes, and whether their want to keep the food costs low. The “Oishi Kenko” application provides users with meal suggestions that simultaneously address these issues as well as ensure nutritional balance. Online nutrition management applications allow users to learn about the food choices they make for their own health. 

To clarify the aspect of the choice behavior of individuals with type 2 diabetes mellitus selecting their meals at home, we evaluated the big data of the food menus selected by individuals in an online diet management application. The saved data included the multidimensional data of the nutrients contained in the menus. We identified specific groups of nutrients in the food menus by clustering the multidimensional data of nutrients after de-dimensioning them and termed these groups as nutritional clusters. We identified the nutritional clusters of food menus that the users with type 2 diabetes mellitus selected and the selection frequency of each nutritional clusters. Then, we comparatively analyzed the constitutional nutrients of the nutritional clusters with the highest and lowest selection frequencies. In this way, we attempted to identify the nutrients in the food menus selected by individuals with type 2 diabetes mellitus at home. Our data can help medical professionals to understand the choice behavior of individuals with type 2 diabetes mellitus at home, sustain self-management in regard with medical nutritional therapy, and improve the quality of life of individuals with type 2 diabetes mellitus.

## 2. Materials and Methods

Individuals who used the online diet management application Oishi Kenko Inc. “Oishi Kenko. Available online: https://oishi-kenko.com/ (accessed on 13 August 2022)” actively from 1 January 2018 to 31 December 2019 were enrolled in the study. This application offers dietitian-directed recipe search and a food menu creation service for the prevention and management of diseases as well as weight loss via web and mobile applications (iOS and Android). Users saved the data of the food menus selected on this application. We performed an observational study and analyzed the saved data on both the web and mobile applications. We removed individual-identifiable information from the data; anonymized active user information, menu preparation information, and recipe search information were exported. 

### 2.1. Definitions

At the start of using the “Oishi Kenko” application, the user checks for type 2 diabetes mellitus, hypertension, and/or dyslipidemia as his or her own attributes. However, the “Oishi Kenko” application does not implement any special suggestions specific to these attributes. The “Oishi Kenko” application allows users to select a single recipe for a dish consisting of specific ingredients. In addition, users can download multiple recipes, such as main and side dishes. The “Oishi Kenko” application aims to fulfill the Dietary Reference Intakes for Japanese (2020 version) proposed by the Ministry of Health, Labor, and Welfare [13]. Users can freely combine recipes developed by registered dietitians and build their own food menus. The “Oishi Kenko” application sums up the nutrients in the food menus constructed by the user and automatically determines whether they meet the Dietary Reference Intakes for Japanese. In this way, when a user builds a food menu using the “Oishi Kenko” application, they can freely select recipes and still build a food menu that meets the Dietary Reference Intakes for Japanese.

Based on this system, the “Oishi Kenko” application claims that users can have a generally healthy diet by using the “Oishi Kenko” application. On the other hand, the “Oishi Kenko” application does not claim specific health benefits such as glucose control, lipid control, weight loss, or blood pressure reduction. In all user attributes, users create food menus by combining recipes that they like and aim to meet the Dietary Reference Intakes for Japanese. In the process of creating these food menus, the app recognizes that certain nutrients are lacking or that certain nutrients are in excess, and the app prioritizes recipes that have the right nutritional balance for that excess or deficiency. In addition, all recipes in the “Oishi Kenko” application are supervised by registered nutritionists for the purpose of balancing nutrition. Therefore, the goal is to create a food menu that is in close proximity to the Dietary Reference Intakes for Japanese, even if the user combines his/her favorite recipes to create a food menu. Based on this premise, “recipes with special nutritional adjustments” are not suggested for user-selected attributes such as type 2 diabetes, hypertension, or dyslipidemia.

All the recipes are annotated with recipe information (annotation data) assigned by a dietitian, and the annotation data related to the ingredients were used in this study. We could not obtain the annotation data from the “Standard Tables of Food Composition in Japan” [14] (Ministry of Education, Culture, Sports, Science, and Technology) for seven “Oishi Kenko” original foods, such as mixtures of barley, rice, and low-protein rice, that are commercially available in arbitrary proportions; therefore, the recipes and menu items containing “Oishi Kenko” original foods were excluded from the analysis.

Information on sucrose is not contained in the Japanese Food Composition Table, which lists 50 nutrient elements in Japanese meals; hence, sucrose was considered the 51st element in this study. The 50 elements in the Japanese Food Composition Table are shown in the Supplement Table. The recipes were annotated with the food composition data of the ingredients. Information on the 50 elements of the Japanese Food Composition Table was recorded as annotation information. Unlike the other 50 elements, for sucrose, the calculations were based on nutrient value data from the Appendix rather than from the main Food Composition Table. The data of the 51 elements were obtained from Python open-source machine learning library StandardScaler in scikit-learn [15]. 

To use the application, the user selects a recipe, creates a menu, and saves it. The saved information is anonymized and stored in the cloud along with the user’s registration information. Since user experience can differ depending on the operating system of a smartphone, the data for iOS only were output in this study as they met the conditions for data analysis.

Users during the study period received an email with instructions on how to use the application, “How to Use Oishi Kenko”, during the first week after signing up. They were not prompted with push notifications on their smartphone devices.

### 2.2. Statistical Methods

Google Colaboratory [16], a Jupyter notebook-based interactive execution environment provided by Google that can run in the cloud, was used, with Python 3.6.9. The files were uploaded to Google Drive, and the optimal number of clusters was studied using the Elbow method “Elbow Method. https://www.scikit-yb.org/en/latest/api/cluster/elbow.html (accessed on 13 August 2022)”. Next, t-distributed Stochastic Neighbor Embedding (t-SNE) was performed with n_components = 2 and perplexity = 50 [17]. The nutrient variables of cluster 5 and 11 were subjected to a *t*-test, and a *p*-value of <0.001 was considered statistically significant. All the data were analyzed using SPSS software (ver. 27; SPSS Inc., Chicago, IL, USA). 

### 2.3. Informed Consent

The users of “Oishi Kenko” are presented with the Terms of Service “Oishi Kenko terms of Service. https://oishi-kenko.com/terms (accessed on 13 August 2022)” and must agree to them before they can start using the application. The service users authorized the use of their information and the data related to their use of this service (including frequency of use, examples of menu preparations, comment status, etc.) for research on health care and agreed to the provision of this information to third parties.

## 3. Results

The food menu data saved in the cloud of the “Oishi Kenko” application were used. In all, 12,201 food menus were saved (Figure 1). Table 1 shows the registration information of the users registered with “Oishi Kenko” as of 19 March 2020. The users were predominantly female, with only 29.3% being male users. The average age of the users was relatively young (41.6 ± 15.2 years), and they were of standard build, with a BMI of 23.4 ± 6.4 kg/m^2^. The prevalence of individuals with type 2 diabetes mellitus was 16.0%, much higher than the prevalence of 8.8% in 2015 among adults aged 20–79 years in Japan. Users with type 2 diabetes mellitus, hypertension, and/or dyslipidemia accounted for 45% of all “Oishi Kenko” users. On the other hand, the most common purpose of use was weight loss (Table 2). 

Of the total 12,201 food menus, 2527 food menus that included “Oishi Kenko” original foods were excluded, and 9674 food menus were included in the analysis. The menus established by the old “menu calendar” (now “meal schedule”) were included (Figure 1). Each food menu was standardized using the 50 elements of the Japanese Food Composition Table and sucrose as explanatory variables. The standardized dataset was subjected to the elbow method to determine the optimal number of clusters, resulting in k = 12 (Appendix A). t-SNE (perplexity = 50) was used to remove dimensions, and the multidimensional data of the menus were displayed in two dimensions (Figure 2). 

Appendix A indicates the number of people in nutritional clusters 0 to 11; the average number of recipes in the food menus; the average age, height, weight, and BMI of the users; and the average number of days per week of app use in 2018–2019 for users in each nutritional cluster. By comparing the frequency of users with type 2 diabetes mellitus in each nutritional cluster, the highest frequency of users was found in nutritional cluster 5 (18.7%), and the lowest, in nutritional cluster 11 (9.5%) (Figure 2). Appendix A indicates there were regular users (more than 4 times a week), occasional users (more than once per week but less than 4 times per week), and very rare users (less than once per week) in nutritional clusters 0 to 11. The proportion of regular users and occasional users was almost the same amongst the clusters. Appendix A indicates the number of times the application was used per week for 12 weeks from the sign-up date. Generally, the number of accesses was high at the beginning of application use but gradually decreased. Appendix A presents the average number of days accessed, days of access to the menus, menus used, menus used per access day, and menus used per day (without or with duplication) during the study period. The actual number of menu references per day was closer when the same menu was accessed and counted as duplicate.

Appendix A shows a comparison of the average number of recipes in food menus, age, BMI, and number of accesses for nutritional clusters 5 and 11; these items were not significantly different between nutritional clusters 5 and 11.

Next, the actual food menus were examined and compared between nutritional cluster 5, which had a high selection frequency among users with type 2 diabetes mellitus, and nutritional cluster 11, which had a low selection frequency. In terms of energy, nutritional cluster 5 had an average caloric value of 149.7 kcal per meal, less than 1/3rd of the 435.2 kcal per meal for nutritional cluster 11 (Table 3). In addition, nutritional cluster 5 had 0.3 μg less vitamin D and 0.1 mg less vitamin B1 than nutritional cluster 11. Meanwhile, nutritional cluster 5 had 1081.8 μg more carotene and 5.3 g more sucrose than nutritional cluster 11. Nutritional cluster 5 more frequently included beef and pumpkin as ingredients, while cluster 11 more frequently included pork and fish. The protein:fat:carbohydrate (often abbreviated “PFC”) ratios of nutritional clusters 5 and 11 were calculated; nutritional cluster 5 had a PFC ratio of 20:33:47%, indicating a low-carbohydrate diet, while nutritional cluster 11 had a PFC ratio of 17:20:63% (Table 4).

Of the 12,201 cases in the total food menu, 2527 cases of the food menu containing”Oishi Kenko” original foods were excluded because the exact calculation of sucrose was not possible due to the inability to cite the Food Composition Table of the Ministry of Education, Culture, Sports, Science, and Technology.

The highest selection frequency was observed for nutritional cluster 5 (18.7%), and the lowest, for nutritional cluster 11 (9.5%). The two-dimensional map shows the relative distances; the legends of the X-axis and Y-axis represent the relative distance.

The basic characteristics of the users registered with the online food management application “Oishi Kenko” are shown.

Users registered their own purpose to use the online food management service ”Oishi Kenko” by themselves.

Table 3 shows the mean and significant differences in nutrients per food menu in nutritional clusters 5 and 11. Nutritional cluster 11 had 149.7 kcal of energy and 47% carbohydrate ratio, significantly different from 435.2 kcal of energy and 63% carbohydrate ratio of cluster 5 (*p* < 0.001, respectively).

Nutritional cluster 5, which had more users with type 2 diabetes mellitus, had a low-carbohydrate diet with a PFC of 20:33:47%, while nutritional cluster 11, which had fewer users with type 2 diabetes mellitus, had a PFC of 17:20:63%.

## 4. Discussion

To our knowledge, the current study is the first study to report data on meal selection from an online meal management app in individuals with type 2 diabetes mellitus and to use big data cluster analysis. In fact, “Medical Big Data in Japan”, a paper published in 2020, does not include dietary content, which is called food menu in this study, although it points out that real-time medical big data, such as medical, health checkup, nursing care, mortality, and lifestyle data, are effective for determining disease trends and appropriate responses [18].

In this study, we used the multidimensional data of the nutrients of food menus selected by users of the online application after de-dimensioning them. The users selected the recipes by looking at their nutritional value and nutritional clusters of staple foods, main dishes, and side dishes in each menu. The frequency of individuals with type 2 diabetes mellitus in each nutritional cluster was compared. The carbohydrate ratio was as low as 47% in nutritional cluster 5, although the Japanese Clinical Practice Guideline for Diabetes 2019 recommend a carbohydrate ratio of 50–60%. The carbohydrate ratio of nutritional cluster 11 was higher than 60%. Interestingly, nutritional cluster 11 had the lowest frequency of selection in users with type 2 diabetes mellitus among the 12 nutritional clusters. Next, we examined the actual food menus of nutritional cluster 5, which had the highest selection frequency in users with type 2 diabetes mellitus. Nutritional cluster 5 was characterized by the lowest carbohydrate ratio, 47%, among all the clusters. In addition, many menus consisted of only a single item. These single-item menus were considered to be the opposite of well-balanced menus that included a staple food, main dish, and side dishes. The average energy of the foods in nutritional cluster 5 was 149.7 kcal. Those who selected food menus from nutritional cluster 5 were considered to be low-carbohydrate and calorie-restricted-diet oriented and weight-loss oriented. In fact, food menus with a carbohydrate ratio of 40% were used as low-carbohydrate diets in several previous studies [19,20]. Since a DIRECT study reported that instructing individuals to limit their carbohydrate intake was found to decrease energy intake and help users to lose weight, individuals seeking to lose weight tend to prefer low-carbohydrate and calorie-restricted diets [19,20,21].

Dietary therapy for diabetes mellitus still plays a central role in the management of blood glucose and the prevention of vascular complications, including nephropathy. In fact, an administrative program to prevent the onset and progression of diabetic nephropathy has been implemented in Japan [22]. A basic concept in the dietary therapy for diabetes is to follow a well-balanced diet with energy-producing nutrients. However, there is not enough evidence on what constitutes a balanced diet. Moreover, individuals with type 2 diabetes need to prepare their own menus at home that consist of well-balanced, energy-producing nutrients without the assistance of a registered dietitian or other health care professionals. To support the dietary therapy for individuals with diabetes mellitus at home, ADA Consensus Statement 2022 and Japanese Clinical Practice Guideline for Diabetes 2019 also recommend nutritional guidance. In Japan, “Food Exchange Lists–Dietary Guidance for Persons with Diabetes, 7th Edition” (edited by the Japan Diabetes Society) is currently used for dietary therapy [23]. Food Exchange Lists are intended to facilitate guidance on complicated dietary regimens and allow individuals to freely prepare menus at home that meet the nutritional requirements and their preferences. However, the lists are becoming complex with each edition, and a limited number of individuals can utilize this information. In addition, some of those who continue to follow the diet based on Food Exchange Lists were reported to be tired of doing so, and some were even reported to lose self-management. There has been some discussion on the decrease in patient satisfaction with continued medical nutritional therapy as directed by medical professionals. The choice behavior of individuals with type 2 diabetes mellitus can help medical professionals to understand these issues.

This study had some limitations. The data obtained were not based on actual menu records, and the medical history and other profiles were self-reported; laboratory data such as blood glucose or HbA1c levels were absent, and the data on pharmacological therapy were also absent. A single user may have selected multiple food menus and used just one. Users of “Oishi Kenko” are motivated to cook meals for themselves or their families in a healthy manner. Therefore, the impact of selection bias could not be ruled out. In addition, the users of “Oishi Kenko” do not use it to cook all their meals. In other words, they use this application when they want to cook healthy food for themselves or their families. Furthermore, they may be eating additional meals besides the ones cooked using this application or they may be snacking more often than needed. However, our results reflect the cooking decisions made by health-conscious individuals with type 2 diabetes mellitus when they want to cook healthy meals, especially low-carbohydrate and energy-restricted meals, for themselves and their families without the intervention of medical professionals. 

The data from this study are valuable because they provide a glimpse of the true feelings of individuals with type 2 diabetes mellitus, which cannot be gleaned from the hospital room. Individuals with type 2 diabetes mellitus who are indifferent to their health tend to stay away from health information [24], and a different approach is needed to understand how they can self-determine their diets without medical intervention.

## 5. Conclusions

Individuals with type 2 diabetes mellitus wish to select low-carbohydrate and low-calorie food menus at home using online diet management applications. To sustain self-management in regard with medical nutritional therapy for diabetes mellitus and to improve the quality of life of individuals with diabetes mellitus, medical professionals could incorporate these choice behaviors into medical nutritional therapy.

## Figures and Tables

**Figure 1 nutrients-14-04290-f001:**
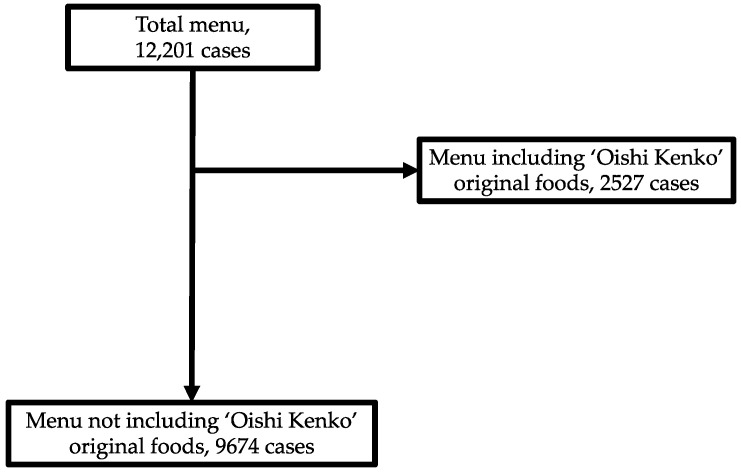
Flow chart of the study datasets.

**Figure 2 nutrients-14-04290-f002:**
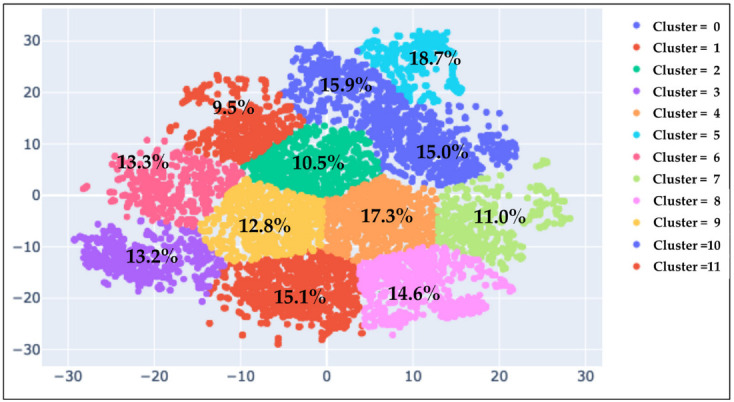
Comparison of the selection frequencies of users with type 2 diabetes mellitus for each nutritional cluster.

**Table 1 nutrients-14-04290-t001:** Basic characteristics of the study population.

Variable	Total
Number of users	72,693
Male (%)	21,274 (29.3%)
Age (yo)	41.6 ± 15.2
Height (m)	1.62 ± 0.09
Weight (kg)	61.4 ± 14.7
Body mass index (kg/m^2^)	23.4 ± 6.4
Number of individuals with type 2 diabetes mellitus (%)	11,625 (16.0%)
Physical activity	
Extremely inactive	39,859 (54.8%)
Sedentary	28,540 (39.3%)
Moderately active	4294 (5.9%)

**Table 2 nutrients-14-04290-t002:** Purposes for using the application of 72,693 users.

Purpose	Number of Users	Frequency
Weight loss	20,904	28.8%
Type 2 diabetes mellitus	13,594	18.7%
Hypertension	11,009	15.1%
Healthy living and disease prevention	10,076	13.9%
Dyslipidemia	7890	10.9%
Acne—Rough skin	2560	3.5%
Pregnancy	2305	3.2%
Anemia	2151	3.0%
Others	2204	3.0%

**Table 3 nutrients-14-04290-t003:** Comparison of nutrients in nutritional clusters 5 and 11.

		Nutritional Cluster 5	Nutritional Cluster 11	*p*-Value
energy	kcal	149.7	435.2	<0.001
water	g	146.2	267.1	<0.001
protein	g	7.5	18	<0.001
amino	g	4.2	13.1	<0.001
fat	g	5.5	9.6	<0.001
triacylglycerol	g	4.3	7.7	<0.001
fatty_acid_sat	g	1.3	2.3	<0.001
fatty_acid_mono	g	2	3.4	<0.001
fatty_acid_poly	g	1.2	2	<0.001
cholesterol	mg	24.1	50.6	<0.001
carbon	g	17.5	66.3	<0.001
carbohydrate_available	g	9.6	58.6	<0.001
fibers_soluble	g	0.6	0.7	<0.001
fibers_insoluble	g	1.7	2.6	<0.001
fibers_total	g	2.6	3.6	<0.001
ash	g	2	3.1	<0.001
sodium	mg	431.4	662.9	<0.001
potassium	mg	384.3	546.1	<0.001
calcium	mg	56.6	66.7	<0.001
magnesium	mg	34.2	58.2	<0.001
phosphorus	mg	100.6	231	<0.001
iron	mg	1	1.5	<0.001
zinc	mg	0.8	2	<0.001
copper	mg	0.1	0.3	<0.001
manganese	mg	0.2	0.8	<0.001
iodine	μg	234.9	285.7	0.256
selenium	μg	4.4	18.3	<0.001
chromium	μg	1.2	1.4	0.015
molybdenum	μg	10.6	49	<0.001
retinol	μg	6.1	13.4	<0.001
carotens_alpha	μg	215.2	98.1	<0.001
carotens_beta	μg	1081.8	528.1	<0.001
beta_crypto	μg	12	5.5	<0.001
beta_caroten	μg	1198.9	581.4	<0.001
retinol_equivalent	μg	105.8	62	<0.001
vitamin_d	μg	0.3	1	<0.001
tocopherols_alpha	mg	1.1	1.2	0.28
tocopherols_beta	mg	0	0.1	<0.001
tocopherols_gamma	mg	1.3	1.3	0.172
tocopherols_delta	mg	0.3	0.2	0.316
vitamin_k	μg	40.6	35.2	0.015
vitamin_b1	mg	0.1	0.3	<0.001
vitamin_b2	mg	0.1	0.2	<0.001
niacin	mg	1.8	5.3	<0.001
vitamin_b6	mg	0.2	0.4	<0.001
vitamin_b12	μg	0.4	1	<0.001
folate	μg	51.3	61.1	<0.001
pantothenic_acid	mg	0.6	1.4	<0.001
biotin	μg	3.3	6	<0.001
vitamin_c	mg	22.6	20.1	0.015
sucrose	g	5.3	3.5	<0.001


**Table 4 nutrients-14-04290-t004:** The protein:fat:carbohydrate ratios of nutritional clusters 5 and 11.

	Protein (%)	Fat (%)	Carbohydrate (%)
Nutritional cluster 5	20	33	47
Nutritional cluster 11	17	20	63

## Data Availability

Data are available upon request owing to restrictions (privacy or ethical).

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
