# Peer review of "Individuals with Type 2 Diabetes Mellitus Tend to Select Low-Carbohydrate, Low-Calorie Food Menus at Home on Diet Application"

_nutrients, 2022, doi:10.3390/nu14204290_

Round 1

Reviewer 1 Report

see attached file

Author Response

Response to Reviewer 1

Comment 1: Tominaga et al. report on a study where they clustered menus using log data from the "Oishi Kenko" application, one of 19 online meal management applications, to identify clusters associated with the highest and the lowest frequency of individuals with diabetes. The research was funded by Oishi Kenko.

Response 1: Thank you for appreciating the value of our paper. As you indicated, this study was funded by “Oishi Kenko” and the authors include employees of “Oishi Kenko”. They contributed to this study by extracting data from the data server of “Oishi Kenko”. On the other hand, they are not involved in the analysis of the data or the writing of the paper. We have clarified this point and added it to the text as follows. We have also attached an affidavit of COI.

****************************************

Page11, Line 327; Conflicts of Interest: All authors declare the following: Tetsuya Nojiri is employed by Oishi Kenko Incorporated. Tetsuya Nojiri contributed to the methodology of this study by outputting data from database of Oishi Kenko Incorporated but were not involved in the analysis and interpretation of the output data or in writing the manuscript.

Kazuki Hamada contributed to the methodology of this study by outputting data from database of Oishi Kenko Incorporated but were not involved in the analysis and interpretation of the output data or in writing the manuscript.

*****************************************

Comment 2: In this study, authors analyzed the big data of the menus selected by people in an online diet management application by de-dimensioning the multidimensional data of the nutrients contained in the menus and analyzing the clusters. By comparing the frequency of patients with diabetes in each cluster, they aim to identify the menu items that people with diabetes prefer to select. This is a welcome and innovative study. Use of big data from app usage could be a mine of information related to health and diseases. The assumption is that if we could know the menus that individuals tend to choose in the real world, this could help to provide targeted and feasible dietary guidance. Results include: Users were predominantly female (29.3% male), relatively young (41.6±15.2 years) and with obesity, with a BMI of 23.4±6.4 kg/m2. The prevalence of diabetes cases was 16%, higher than the prevalence of 8.8% reported in 2015 in adults aged 20-79 years in Japan. The total number of menu items was 12201, and 9674 were included in analysis. Oishi Kenko ‘original food’ was excluded as composition could not be ascertain. Each menu was standardized using the 50 elements of the Japanese Food Composition Table plus sucrose. The highest number of people with diabetes was found in Cluster 5 (18.7%) and the lowest in Cluster 11 (9.5%).  The cluster with the highest frequency of people with diabetes had 149.7 kcal of energy and 47% of carbohydrate, different from the cluster with the lowest frequency of 435.2 kcal of energy with 63% carbohydrate.

While the data are interesting, there are elements that are missing, which limit the overall impact of the results and their meaningfulness.

Response 2: Thank you for summarizing the results of our paper. We have revised the paper according to your suggestions.

Comment 3: The data are based on self-selected choice of menus/ recipe and in no way represents what people/ family actually eat. The person using the app may cook for someone else- or an entire family- and, if the menus are actually eaten, there is no guarantee that the app user is eating it. So, at best, the analysis allows to identify wishful behavior on the part of the users. In a similar vein, users may browse an app to look for sport’s clothing or equipment but never actually buy them and/or wear them.

Response 3: Thank you very much for your very valuable suggestions! We revised title abstract, the introduction section, and the conclusion section and indicated that 'To clarify wishful behavior on the part of individuals with diabetes mellitus at home' is the Goal of the study.

Comment 4: The second main limitation is whether these data are representative of the diet of the app users. Authors need to provide information about the number of menu items selected per participant and over which period of time. Table 2 reports n= 72 693 and Methods report that 9674 menu items included in the analysis. This represents less than 0.14 menu item per participant. Is this correct? And these data were collected over how many weeks/months? One user may have used the app sporadically, once a month, while another has used it more or even less frequently, and consistently. These data are essential to see how representative it is from habitual diet choices. Frequency, consistency and duration of app usage for the entire cohort are key to validate their data.

Response 4: We apologize that our representation was misleading; some of the N's refer to the number of users of the application and some to the number of menus registered on the server. We have first corrected the text in this regard and no longer use the small abbreviation N.

the number of users is 72693.

The number of menu items and the number of users who created menu items in each cluster are shown (Table S1).

Comment 5: Thirdly, please provide frequency, consistency and duration of app usage for the 2 clusters with high and low % of individuals with diabetes. Duration of usage of the app and number of menus per participant is essential for data to be meaningful.

Response 5: Thank you for your valuable comment. We added Table S1-2 which reveal the frequency of application use per week.

Comment 6: Fourthly, please give more details about the Oishi Kenko original food:  are these commercial items sold through the app? While the rationale of why they are excluded is well explained in legend of Figure 1, it is puzzling that these data were not included. At the very least, please provide the proportion of original food as part of the diet for the entire sample, and in each of the 2 selected clusters. Does this original food target weight loss? Or any other health condition? Do they make any health claim?

Response 6: We apologize for any inadequacies in our explanation for the Oishi Kenko original foods. We have added the following sentence in response to the reviewer's suggestion to help the reader understand the Oishi Kenko original foods.

****************************************

Page3, Line 97; The "Oishi Kenko" application allows users to select a single recipe for a dish consisting of ingredients. In addition, users can record multiple recipes, such as main and side dishes, as a menu. All recipes are annotated with recipe assignment information (annotation data) by the "Oishi Kenko" dietitian, and the annotation data related to the ingredients were used in this study. Seven foods such as mixtures of barley rice and low-protein rice are mixtures of commercially available foods in arbitrary proportions, and these are referred to as the “Oishi Kenko” original foods. It is impossible to obtain annotation data of the “Oishi Kenko” original foods from the Food Composition Table of the Ministry of Education, Culture, Sports, Science and Technology. For this reason, recipes containing the “Oishi Kenko” original foods and menu items containing them were excluded from the analysis.

****************************************

Comment 7: What does ‘diet’ mean on Line 143 (and Table 2) :’ On the other hand, the most common purpose of use was diet’? was it weight loss?

Response 7: Thank you for pointing out. The term ‘diet’ was academically inappropriate. We will correct it to ‘weight loss’.

Comment 8: Figure 2: the percentages are not aligned with each color, confusing to read. It looks like cluster 5 and 11 are next to each other, but this is not clear

Response 8: Thank you for your suggestion. We have adjusted the size of the cluster percentage numbers. The color tone is not changeable.

Comment 9: In Table 1 and 2, add and compare characteristics of subjects in the 2 chosen clusters: age, BMI, desire to diet, etc..

Response 9: Thank you for your suggestion. We added Table S1-2 which reveal the profile of each cluster.

Comment 10: Label of table 3 needs more details. Are the kcal and composition described by meal unit? Menu? In table 3, please add a column for each cluster, adjusting each ingredient for calories amount and compare the 2 groups. Explain the color code in words as foot note.

Response 10: Thank you for your suggestion. We revised the title of table 3 and described legend of table 3 as follows; Table 3 shows the mean and significant differences in nutrients per menu in clusters 5 and 11. The frequency of individuals with diabetes is the highest in Cluster 5 (18.7%) and is the lowest in Cluster 11 (9.5%). Cluster 11 had 149.7 kcal of energy and 47% carbohydrate ratio, significantly different from the cluster with the lowest frequency of 435.2 kcal of energy and 63% carbohydrate ratio (P < 0.001, respectively).

Comment 11: Please clarify how many participants and how many meal/menus per participants, including range.

Response 11: Thank you for your suggestion. We added Table S1-2 which reveal the profile of each cluster.

Comment 12: Single item versus full menu: please clarify in methods and results/ tables which data is which.

Response 12: Thank you for your suggestion. We added Table S1-2 which reveal the profile of each cluster. On average, each menu consisted of about 2.8 items.

Comment 13: Please use person first language: not ‘diabetics’, but ‘individuals with diabetes’.

Response 13: Our apologies for our lack of consideration. We revised them as ‘individuals with diabetes’.

Reviewer 2 Report

The study about food choice of people is interesting. However, I am not sure what is new in the paper from a science point of view. People with diabetes tend to choose a low energy and low-carbohydrate food.--- What is new there? Or the authors are trying to say we should have a choice system where participants can select their food with the guidance from a nutritionist?

1). The paper is difficult to understand, both in writing and contents. For example, cluster-- not sure what they mean, "real world"-- there is a virtual world in the paper trying to compare? Maybe just describe what your data is: people who chosen xxx food..

2). The goal was to identify what most people with diabetes are eating? Nothing wrong there, but need to get into a little more detailed in the text.  and may be recommend improvement? 

3). The discussion needs to explain the results and comparing with results from other studies and extend on recommendation of the results....

Author Response

Response to Reviewer 2

Comment 1: The study about food choice of people is interesting. However, I am not sure what is new in the paper from a science point of view. People with diabetes tend to choose a low energy and low-carbohydrate food.--- What is new there? Or the authors are trying to say we should have a choice system where participants can select their food with the guidance from a nutritionist?

Response 1: We appreciate your suggestions to enhance the novelty of this paper. We revised introduction section and indicated that 'To clarify wishful behavior on the part of individuals with diabetes mellitus at home' is the Goal of the study.

We specify that the novelty of our study is that our analysis allows us to identify wishful behavior on the part of users. Related to this point, we agree that the meaning of the term "real world" was unclear, as you pointed out.

****************************************

Page 1, Line 42; However, the physician-patient relationship of dietary therapy is also controversial. The decrease in patient satisfaction with continuing the diet therapy as directed by medical professionals has also been debated. The possibility of burnout from continuing the diet therapy as directed by medical professionals has been also discussed.6 We considered the possibility that medical professionals could reduce burnout and increase quality of life by implementing the dietary therapy that takes into account wishful behavior on the part of individuals with diabetes mellitus.

To clarify wishful behavior, research methods in which medical professionals collect information about food selection from patients in a hospital setting are undesirable. In fact, Matsumoto et al. reported in a home blood pressure survey that blood pressure readings unrecognized and recorded by participants were higher than those self-reported by patients to their health care providers. Thus, we are concerned that hospital surveys of food selection at home may diverge from wishful behavior on the part of individuals with diabetes mellitus at home.

****************************************

Page 2, Line 70; To clarify wishful behavior on the part of individuals with diabetes mellitus at home, we evaluate the big data of the food menus selected by individuals in an online diet management application. The saved data of online diet management application is consisted with multidimensional data of the nutrients contained in the menus. We identify some groups of the food menus, which is called nutritional clusters of food menu, by clustering the de-dimensioned multidimensional data of the nutrients. We identify the nutritional clusters of food menu which individuals with diabetes mellitus prefer to select by comparing the frequency of individuals with diabetes mellitus in nutritional clusters of food menu. Following that, we analyzed the constitutional nutrients of the nutritional cluster with the highest frequency of individuals with diabetes comparing with the nutritional cluster with the lowest frequency. In this way, we thought it would be possible to identify the nutrients of food selection by individuals with diabetes mellitus at home. This will help medical professionals to know wishful behavior on the part of individuals with diabetes mellitus at home, reduce from continuing the diet therapy, and improve quality of life of individuals with diabetes.

****************************************

Comment 2: The paper is difficult to understand, both in writing and contents. For example, cluster-- not sure what they mean, "real world"-- there is a virtual world in the paper trying to compare? Maybe just describe what your data is: people who chosen xxx food..

Response 2: Thank you for pointing this out. I agree with the reviewer that revising the text to make the content more understandable is important to the value of this paper.

We added explanation for Cluster as follows; We identify some groups of the food menus, which is called nutritional clusters of food menu, by clustering the de-dimensioned multidimensional data of the nutrients (Page 2, Line 74).

We used the term real-world information in this paper not to refer to diet-oriented information of people with diabetes collected by medical professionals, but rather to diet-oriented information of application users, including people with diabetes, that is not collected by medical professionals. We have dropped the use of the term real-world because, as you pointed out, it is misleading and does not accurately convey our intent above.

We revised the term logs as follows; the saved data of the selected food menus in online diet management application (Page2, Line 72). The term menu was revised ad food menu.

Comment 3: The goal was to identify what most people with diabetes are eating? Nothing wrong there, but need to get into a little more detailed in the text.  and may be recommend improvement?

Response 3: I agree with the reviewer's point that we should be clear about the goal of this paper. We revised title abstract, the introduction section, and the conclusion section and indicated that 'To clarify wishful behavior on the part of individuals with diabetes mellitus at home' is the Goal of the study.

Comment 4: The discussion needs to explain the results and comparing with results from other studies and extend on recommendation of the results....

Response 4: In accordance with your comments 1-3, we clarified in the preface that the purpose of our study is to identify the nutrients in the desired dietary choices made by people with diabetes at home.

In order to achieve this objective, it is not desirable to conduct the study in a hospital setting, where medical professionals collect information from patients about their food selection. In fact, Matsumoto et al. reported in a home blood pressure survey that blood pressure readings unrecognized and recorded by participants were higher than those self-reported by patients to their health care providers. Thus, we are concerned that hospital surveys of food selection at home may diverge from wishful behavior on the part of individuals with diabetes mellitus at home. For this reason, we described that to achieve our objective, we decided to collect large-scale data stored in an online dietary management application and group the data according to nutrients, comparing groups with a higher frequency of people with diabetes with groups with a lower frequency of people with diabetes.

Related to this, the importance of diet in diabetes care, which was originally described in the Introduction, has been moved to the Discussion section. We also noted the significance of our analysis in allowing health care providers to modify nutritional guidance based on the dietary nutrient choices made by people with diabetes in their daily lives, taking into account their true desires. In other words, the diet is concerned with the burnout and quality of life of people with diabetes. The possibility of reducing burnout and quality of life is mentioned if health care providers can modify nutritional guidance to take into account the preferences of people with diabetes.

Round 2

Reviewer 1 Report

The authors have not responded with details and precision to the comments .

the responses should include the response, not just referring to the  supplementary tables.

The supplementary tables are not accessible.

Reviewer 2 Report

The manuscript is still very difficult to comprehend because of the grammatic and English issue. They really need to re-work the paper.  The topic is somewhat interesting, but will need outside helpers to re-organize the data, goal and design. 

Author Response

Response to Reviewer 2 (Round 2)

Comment 1:The manuscript is still very difficult to comprehend because of the grammatic and English issue. They really need to re-work the paper.  The topic is somewhat interesting, but will need outside helpers to re-organize the data, goal and design. 

Response 1: We asked for a professional native English speaker to solve the grammatic and English issue and to revise the manuscript again. The data, goal and design have also been revised. 

Round 3

Reviewer 1 Report

While the authors have provided answers, some are still missing and the paper still requires clarification as well as language edits.

1-It is still very difficult from the data presented to figure out frequency of app usage: whether there was some regular users, occasional users, or very rare users [all of these will need to be defined], and whether the proportion of regular users and occasional users was the same amongst the clusters.

2- The duration of data collection is also unclear. Please specify number of months (with dates) when data were collected.

3- What was the pattern of the usage of the app ? people using it frequently after signing up for the app then less frequently, then not using it at all? Did the usage of the app increase, decrease, remain the same during the entire study period?

During the study period, between 2018 and 2019 [please provide exact date intervals], how many days each user used the app? 

4- Do app users receive prompts to remind them to use the app? If they do, how often and which format (text messages, email ?)

5-Why is the ‘average number of days of app use per week in 2018-2019’ different for clusters 5 and 11 between table S1 and S2?

6- On days when people use the app, how may menus did they use on average? Does the column ‘average of menu count’ refers to the number of menus use on days they use the app? 

7- What is the minimum (total number of days over the entire study period, and total number of menus looked at over the study period, and number of menus/ day) and the maximum usage of the app reported? 

Is the study period 12 months?

8- Please clarify whether the Oishi Kenko app claims any health benefits, weight loss, glucose control or other health benefits to their users? And whether these claims are targeting certain groups of users, ie the one who report they have diabetes or a high BMI?

9- The title does not need to use the exact words of the reviewer.

Suggest instead: ‘Individuals with diabetes mellitus tend to select low-carbohydrate, low calorie food menu from an online nutrition management system.’

10- Objective:

‘To clarify the role of wishful behavior when individuals with diabetes mellitus select their meals at home, we evaluated the big data of the food menus selected by the individuals on an online diet management application.’ 

Suggest instead: To assess menu selection from an online diet management application by individuals with diabetes over a period of xx [add duration of data collection here] months.

11-In Discussion:

To our knowledge, the current study is the first study to report the wishful behavior of individuals with diabetes mellitus when selecting meals, which was analyzed using big data of the food menus selected by the users on the online diet management application. 

Suggest replacing with:

To our knowledge, the current study is the first study to report data on meal selection from an online meal management app by individuals with diabetes mellitus, and to use big data cluster analysis.
